# Causes and Phenotypes of Work-Related Asthma

**DOI:** 10.3390/ijerph17134713

**Published:** 2020-06-30

**Authors:** Piero Maestrelli, Paul K. Henneberger, Susan Tarlo, Paola Mason, Piera Boschetto

**Affiliations:** 1Department of Cardiac-Thoracic-Vascular Sciences and Public Health, University of Padova, 35128 Padova, Italy; paola.mason.1@unipd.it; 2Respiratory Health Division, National Institute for Occupational Safety and Health, Centers for Disease Control and Prevention, Morgantown, WV 26505-2888, USA; pkh0@cdc.gov; 3Toronto Western Hospital, University Health Network, St. Michael’s Hospital and University of Toronto, Toronto, ON M5G 2P1, Canada; susan.tarlo@utoronto.ca; 4Department of Medical Sciences, University of Ferrara, 44121 Ferrara, Italy; bsp@unife.it

**Keywords:** occupational disease, exposure, workplace, prevention, respiratory system

## Abstract

Work-related asthma (WRA) includes heterogeneous conditions, which have in common (i) symptoms and signs compatible with asthma and (ii) a relationship with exposures in the workplace. The types of WRA described in this review are distinguished by their etiology, comprising of work-exacerbated asthma (WEA), irritant-induced asthma (IIA), and immunologic occupational asthma (OA). There have been significant advances in the definition and characterization of the different forms of WRA by international panels of experts. The present review provides a comprehensive and updated view of the current knowledge on causes and phenotypes of WRA. Health care practitioners should consider WRA in any case of adult asthma, given that one fifth of workers with asthma report symptoms of WEA and it has been estimated that OA represents 10% to 25% of asthma in adulthood. The information provided in this review will facilitate the physician in the recognition of the different forms of WRA, since it has been established that five categories of agents are responsible for at least 60% of WEA cases and seven groups of agents are the cause of 70% of immunologic OA. In addition, there is agreement that IIA can be elicited not only by a single massive irritant exposure, but also by low/moderate repeated irritant exposures.

## 1. Introduction

The term work-related asthma (WRA) is an umbrella diagnosis that covers heterogeneous conditions, which have in common (i) symptoms and signs compatible with asthma and (ii) a relationship with exposures in the workplace. The types of WRA described in this review are distinguished by their etiology, comprising work-exacerbated asthma (WEA), irritant-induced asthma (IIA), and immunologic occupational asthma (OA). Within each of these conditions, further levels of heterogeneity, i.e., more phenotypes, are recognized.

There have been significant advances in the definition and characterization of the different forms of work-related asthma by international panels of experts. Regarding WEA, the case definition criteria, prevalence, most important causes, and socioeconomic impact have been established. Regarding IIA, there has been consensus on its definition and probability, and on IIA phenotypes due to repeated irritant exposures at work, in addition to the most common recognized diagnosis that coincides with the reactive airway dysfunction syndrome (RADS). The largest ever survey conducted by the European network on Phenotyping of Occupational Asthma (E-PHOCAS) identified the major causal agents of immunologic OA and demonstrated the heterogeneity of the diseases. The present review takes advantage of the expertise of the investigators who participated in those panels to give a comprehensive and updated view of the current knowledge on causes and phenotypes of work-related asthma.

## 2. Work-Exacerbated Asthma (WEA)

The short definition of work-exacerbated asthma (WEA) presented in the 2011 American Thoracic Society (ATS) statement on WEA is “worsening of asthma due to conditions at work” [1]. The case definition has four criteria. First, (i) asthma onset occurs either before starting in the worksite of interest (i.e., pre-existing asthma) or while employed in that worksite, but is not attributed to workplace conditions suspected to cause exacerbation (i.e., concurrent or coincident asthma). Two other criteria are that (ii) conditions are present in the workplace that could exacerbate asthma and (iii) the worker experiences those conditions before his asthma worsens. Finally, (iv) it is unlikely that the response to workplace conditions is occupational asthma (OA), which is asthma caused by work.

The 2011 ATS statement reported a summary WEA prevalence of 21.5%, which was the median of estimates reported by 12 studies [1]. These studies, conducted in the general population or general healthcare settings, determined WEA status on a case-by-case basis and expressed prevalence as the percentage of either all adults with asthma or working adults with asthma. Five of the studies were from the US and the other seven studies were conducted in six other countries (i.e., Australia, Brazil, Canada, Finland, Singapore, and Sweden). Updating the literature search from the 2011 ATS statement to March 2020 identified an additional six studies with eight estimates of WEA prevalence. Several of these studies were conducted in Europe by surveying adults with asthma identified via healthcare records in Great Britain [2], Italy [3], and Spain [4]. A fourth study was based on the European Community Respiratory Health Survey (ECRHS) and used data from 25 study centers in 11 European countries and one center in the US [5]. Participants had been selected either at random or because they had reported respiratory symptoms during the first round of the ECRHS (1991–1993) and had also taken part in the second round (1998–2003). The final two studies, conducted in the US, were based on the adult Asthma Call-Back Survey of the Behavioral Risk Factor Surveillance System. One study used data collected in 2005 and provided separate prevalence estimates for the states of Michigan (MI), Minnesota, and Oregon [6] and the other used 2006 and 2007 data from New York (NY) [7]. The median WEA prevalence for the combination of the more recent studies and the prior 12 was 21.5%, the same as reported in 2011. A potential limitation of most of these studies was an inclusive definition of WEA based on a self-reported association of asthma symptoms with work. Three studies referenced in the 2011 ATS statement used more objective criteria for WEA and had a lower median prevalence estimate of 14% [1]. One of the more recent studies applied more objective criteria for WEA and reported a 15% prevalence [4]. The four studies considered together had a median estimate of 14.5%. The 2011 ATS Statement on WEA observed that many different workplace conditions can exacerbate asthma [1]. A similar diversity is evident in subsequent articles and reports. For example, US state-based surveillance programs have recorded up to three putative agents for each WEA case. The National Institute for Occupational Safety and Health (NIOSH) has summarized and posted exposure data online for these cases identified in the states of California (CA), Massachusetts (MA), MI, and New Jersey during 1993–2006 (1307 WEA cases, 1925 agents) and in the states of CA, MA, MI, and NY during 2009–2012 (902 WEA cases, 1328 agents) [8]. The earlier data have been removed from the internet and are available from the Surveillance Branch of the Respiratory Health Division at NIOSH. Table 1 presents the distribution of these 3253 agents associated with the 2209 surveillance WEA cases, with separate entries for the 14 categories that had at least 50 reports and a final category for all other agents. The top three categories account for almost half of the agents (44.3%) and are miscellaneous chemicals and materials (including pesticides and glues), mineral and inorganic dusts, and cleaning materials. Irritant compounds are prominent in some categories, such as acids, bases, and oxidizing agents and cleaning materials. Agents that are not unique to workplaces are listed in several categories, including cigarette smoke and diesel exhaust in pyrolysis products, common allergens such as flour and pollen in plant materials, and cats and animal dander in animal materials. Finally, there are conditions that are traditional triggers of asthma symptoms, such as exercise and stress in the ergonomics category and cold and hot temperatures in the physical factors category.

Similar to findings from surveillance in the US, risk-set studies conducted in Europe have identified a variety of workplace exposures associated with the exacerbation of asthma. In these studies, researchers modelled exacerbation among working adults with asthma and tested whether workplace exposures assessed by an asthma-specific job exposure matrix (JEM) or self-reports were associated with exacerbation while controlling for potential confounders. From the ECRHS, severe exacerbation of asthma was associated with different JEM-assigned exposures: high dust, gas, or fumes; high gas and fumes; high mineral dust; and both low and high biological dust [5]. Another study used data from five existing investigations conducted in Sweden, modeled three levels of exacerbation (i.e., mild, moderate, and severe), and assessed exposure with both self-reports and a JEM [9]. Severe exacerbation of asthma was associated with self-reported gas, smoke, or dust; organic dust; dampness and mold; cold conditions; and physically strenuous work. Using exposures assigned by a JEM, mild exacerbation was associated with low-molecular weight agents and any asthmagen.

The 2011 ATS statement on WEA summarized data from several studies and concluded that WEA and OA cases were similar regarding the frequency of employment and income loss and that in most studies, changes in job or employer were less common for WEA than OA [1]. Subsequent studies that followed-up WEA and OA tertiary clinic cases in the Canadian provinces of Ontario [10] and Quebec [11] reported on some of the same comparisons. In confirmation of the earlier observations, both studies reported that patients with WEA and OA were equally likely to be employed at follow-up and the Quebec study found that job changes were less likely with WEA (42%) than OA (77%). Even after adjusting for potential confounders, the WEA patients in Quebec were more likely to have kept a job with the same employer than their OA counterparts [11]. However, the results for income loss differed from those presented in the ATS statement. Specifically, income loss was less common for WEA than OA in the Ontario study [10] and any change in income of at least $5000 was less common for WEA in the Quebec study [11]. Results for income loss in the ATS statement were from studies conducted in the United Kingdom and Belgium and showed little difference between the two types of cases. The contrast with findings from Canada could be influenced by country-specific compensation and employment law differences.

In conclusion, WEA is common, with median prevalence estimates of 21.5% among studies with WEA case definitions based on self-reports and 14.5% with more objective definitions. Consistent with the relatively high prevalence estimates, a variety of workplace exposures can exacerbate asthma. Depending on the specific outcome, WEA cases experience socioeconomic consequences that are either equivalent to or less severe than those of OA cases.

## 3. Irritant-Induced Asthma (IIA)

Irritant-induced asthma (IIA) is the term used to describe asthma caused by exposure at work to substances that cause asthma through a lower airway irritant mechanism rather than by immunologic sensitization. As reviewed in 2014 in an European Academy Allergy Clinical Immunology (EAACI) Position Paper [12], the most definitive diagnosis of IIA meets the criteria of reactive airways dysfunction syndrome (RADS) [13] with the new onset of asthma symptoms, persisting for at least 3 months, within 24 h after a single high level exposure to expected irritating gas, fumes, smoke, or vapor in the absence of preceding lung disease, and a significant bronchodilator response on spirometry or positive methacholine challenge. High-level dust (e.g., the exposure of first responders at the New York World Trade Center collapse) has also been included as a cause for definite IIA [14]. Additionally, phenotypes are probable IIA (later onset, after 24 h, or more insidious onset and/or one or more exposures) and possible IIA (associated with chronic moderate irritant exposures) [12] as detailed in the Position Paper with examples of causes. This section will provide an update based mainly on publications since the 2014 EAACI paper.

Additional studies have supported the concept that low/moderate chronic irritant exposures may cause asthma. Dumas and colleagues [15] surveyed over 50,000 adults in Estonia with a questionnaire and assessed work exposures with an asthma-specific job-exposure matrix. Exposure to low/moderate levels of respiratory irritants was reported by 17.4% and was significantly associated with doctor-diagnosed asthma (OR 1.88, 95% CI 1.48–2.37).

It has been well recognized that exposure of professional cleaners and others to cleaning and disinfecting agents can result in the development of IIA from accidental high level exposures [16,17]. However, lower exposures also have been associated with increased risk of asthma (as reviewed by Folletti et al. for studies up to 2016 [18]). Workers included in the epidemiologic reports of cleaners may include a combination of those with OA from sensitization or work-exacerbated asthma as well as definite, probable, and possible IIA [12]. Sensitization causing asthma may result from enzymes or quaternary ammonium compounds and non-occupational asthma can be work-exacerbated in cleaners from exposure to common allergens (for domestic cleaners) or from irritant effects of cleaning agents in workers with hyperresponsive airways [16]. Therefore, it is difficult to draw firm conclusions from these studies.

Three recent questionnaire studies from a single research group assessed asthma among healthcare workers. A cross-sectional study that included housekeepers and medical instrument cleaners found that use of various combinations of alcohol, bleach, high level disinfectants, and enzymes were associated with questionnaire-based clusters of undiagnosed asthma and asthma exacerbations [19]. In contrast, a 6-year prospective cohort study of over 60,000 nurses at a late stage of their career found no significant association between exposure to disinfectants and observed asthma incidence, but questioned whether this may have resulted from a healthy worker effect [20]. However, from another grouping drawn from the same larger cohort of female nurses, the same group found an association between questionnaire reported doctor-diagnosed chronic obstructive pulmonary disease (COPD) incidence and more frequent use of disinfectants, once a week or more [21]. In the absence of clinical investigation results including pulmonary function, it is unclear whether or not these individuals may have had asthma or asthma-COPD overlap rather than COPD.

A review of cases of respiratory disease included 6% of non-asbestos causes that were attributed by physicians to cleaning agents and reported to The Health and Occupation Research (THOR) surveillance network in England, 1989–2017. Chlorine or chlorine-releasers were identified in 26% of the implicated cleaning agents, with a small percentage, less than 4%, attributed to chloramines and nitrogen trichloride. Other potential irritants implicated less frequently than chlorine included acids and caustic agents [17]. The main occupational group were cleaners, followed by nurses/nursing assistants. Among respiratory cases attributed to cleaning agents, 27% occurred from spills. Sixty percent of the 690 cases reported to the Surveillance of Work-Related and Occupational Respiratory Disease (SWORD) system resulted in a diagnosis of asthma, but clinical details were not given and only 2 cases were diagnosed as RADS [17]. Further support for a possible role of chlorine/chlorine releasers was provided from an assessment of the frequency of use of bleach among 607 women in the Epidemiological Study on the Genetics and Environment of Asthma (EGEA) study [22]. Bleach use was significantly associated with adult-onset non-allergic asthma, adjusted OR 4.9; 95%CI 2.0–11.6. Other positive associations were seen with blood neutrophil counts and in non-allergic women with bronchial hyperresponsiveness, cough, and asthma-like symptoms [22].

Irritant chlorinated products producing asthma have been reported as a result of interaction between chlorinated cleaning products and urine, as may occur in a healthcare setting [23]. In an outbreak of cough, dyspnea, upper respiratory symptoms, and airway hyperresponsiveness among several members of a swim club at an indoor pool associated with increases in air levels of trichloramine, a polyamine glue was thought to be the source of nitrogen that reacted with chlorine from the pool to produce the irritant chloramine [24].

Potentially, any irritant respiratory exposure may cause IIA. An evidence-based review by Baur [25] indicated moderate evidence for the following as causes of IIA (although some on this list may also cause sensitization): benzene-1,2,4-tricarboxylic acid, 1,2-anhydride (trimellitic anhydride), chlorine, cobalt, cement, environmental tobacco smoke, grain, welding fumes, construction work, swine or poultry confinement or farming, and the 2001 World Trade Center disaster exposure.

Hypothesized mechanisms of IIA include neurogenic inflammation and oxidative stress [12]. Transient receptor potential channels may have a role in activating some of the responses [26]. In addition, some irritant agents such as ozone and nitrogen oxides can enhance the airway response to allergens [27,28]. Genetic factors have also been suggested to play a role in IIA as indicated by the finding of genetic polymorphisms with genes that play a role in inflammation via the NF-kappaB pathway [29]. Typically, the airway cellular response to irritants is neutrophilic [30].

In conclusion, diagnostic difficulties remain for phenotypes of IIA that do not meet the initial criteria for RADS. Although the level of diagnostic certainty does not generally affect clinical management, it is important for determining the existence of these and for potential workers’ compensation decisions. There is a need for further detailed clinical reports and objective markers to clarify these diagnoses.

## 4. Immunologic Occupational Asthma (OA)

Causative agents of OA can be separated into high-molecular weight (HMW) and low-molecular weight (LMW) agents. It is generally accepted that the first ones act through type I hypersensitivity mechanisms and cause the production of specific IgE antibodies. On the contrary, few LMW sensitizers induce a documented Ig-E mediated response (e.g., platinum, nickel, and chrome salts, reactive dyes, acid anhydrides) [31], suggesting that other mechanisms, not yet characterized, are involved in the respiratory sensitization. Differences in the pathogenesis of immunological OA might explain differences in clinical presentation that some previous studies have evidenced.

Recently, the European network on Phenotyping of Occupational Asthma (E-PHOCAS) conducted an international, multicentre, retrospective study [32] in a large cohort of subjects with OA documented by a positive specific inhalation challenge (SIC). The aim of the study was to compare a wide spectrum of clinical, functional, and inflammatory characteristics of subjects with OA due to HMW and LMW agents and determine whether these two categories of sensitizers are associated with different phenotypes. The cohort included 635 patients with OA induced by LMW compounds (mostly isocyanates) and 544 patients with OA induced by HMW agents (mostly flour). It is remarkable to note that only 8 agents are responsible in more than 70% of the cases of OA (Figure 1). The LMW group showed more symptoms at work (chest tightness and daily sputum), a higher risk of severe asthma exacerbations, and late asthmatic reactions to SIC. The HMW group exhibited more work-related conjunctivitis and rhinitis and more baseline airflow limitation, more atopy, early asthmatic reactions, higher levels of fractional exhaled nitric oxide (FeNO), and blood eosinophils.

The findings of this study are consistent with those of previous smaller studies that analysed atopy, rhinitis, conjunctivitis, and the type of asthmatic reactions in OA induced by LMW and HMW agents [33,34,35]. Moreover, the latency period and the duration of symptomatic exposure were significantly shorter in LMW sensitized patients. This is discordant from previous monocentric studies, which did not detect any differences in latency period and duration of symptomatic exposure [35,36] among HMW and LMW sensitized patients. A plausible hypothesis for the E-PHOCAS findings is that the shorter length of time necessary for sensitization to LMW agents may depend on their workplace concentrations and on the nature of each agent. This first E-PHOCAS analysis also documented a higher rate of severe exacerbations in OA due to LMW agents independently from global assessments of asthma severity grade according to either the European Respiratory Society/American Thoracic Society (ERS/ATS) criteria [37] or the Global Initiative on Asthma (GINA) treatment steps (https://ginasthma.org). At the present time, there is scarce information on the relevance and causes of severe OA. Meca et al. [36] analysed a monocentric group of patients with OA and detected a higher risk of severe asthma graded by GINA guidelines 2010 in OA caused by LMW agents.

More recently [38], the E-PHOCAS network characterized the burden of severe OA in the multicentre cohort of SIC-positive patients (*n* = 997). Severe asthma was defined according to ERS/ATS criteria by a high level of treatment and any of the following criteria: (1) daily need for a reliever medication, (2) 2 or more severe exacerbations in the previous year, or (3) airflow obstruction. A relevant proportion of subjects with OA experienced severe OA (16.2%). This estimate is higher than that described for the adult common asthma in a previous study, which applied the same definition of severe asthma [39] (6.3%). The E-PHOCAS results are in accordance with the findings of Lemiere et al. [40], who demonstrated that OA is associated with a higher risk of severe exacerbations requiring an emergency room visit or hospitalization and greater use of health-care services than common asthma. In the E-PHOCAS study, severe OA was associated with persistent exposure to an inciting agent and a longer duration of the disease. Intriguingly, it was also associated with childhood asthma, a low level of education, and sputum production, as documented in studies in adult common asthma [41,42,43]. As previously described by Descatha et al. [44], severe OA at diagnosis was not associated with the type of causal agent. Anyway, this point should be examined in further longitudinal studies because it is not consistent with the results of previous follow-up studies of OA [45,46] that documented a worse outcome of OA due to HMW compounds even though exposure was ceased.

In conclusion, these recent publications contribute to demonstrate that the term asthma is an umbrella diagnosis for different diseases with different presenting symptoms and grades of severity applicable to immunologic OA also. Mechanistic pathways underlying this phenotypic heterogeneity are still mostly unclear.

## 5. Conclusions

WRA is an important issue in public health since it has the risk of long-term impairment and adverse socioeconomic effects, but it is potentially preventable. Clinical, functional, and pathological alterations in WRA are similar to those found in common asthma, therefore the way to identify this condition is the assessment of workplace exposures and questioning any history of worsening with exposure. Further objective tests should be performed for diagnosis. Health care practitioners should consider WRA in any case of adult asthma, given that one fifth of workers with asthma have a history of WEA and OA represents 10% to 20% of asthma in adulthood. Prompt and correct recognition of these conditions are essential to guarantee an adequate treatment of the disease and prevention of its adverse effects. Indeed, management differs in the case of WEA, IIA, or immunologic OA. A limitation of the international consensus on the phenotypes of WRA, presented in this review, is that it is based on heterogeneous literature, which generally does not meet the criteria of high quality research, like a randomized controlled trial design, as often happens in the field of occupational medicine. Moreover, the mechanisms by which the occupational agents elicit or exacerbate WRA are poorly understood and workplace exposure is assessed by JEM or self-reports rather than quantitative measurements in most of the studies. However, we are confident that the information provided in this review will facilitate the physician in the recognition of the different forms of WRA, since it has been established that five categories of agents are responsible for at least 60% of WEA cases (Table 1) and seven groups of agents are the cause of 70% of immunologic OA (Figure 1). In addition, there is agreement that IIA can be elicited not only by a single massive irritant exposure, but also by low/moderate repeated irritant exposure.

## Figures and Tables

**Figure 1 ijerph-17-04713-f001:**
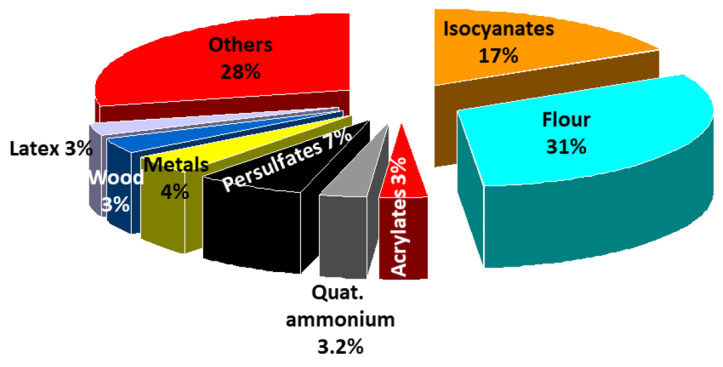
Distribution of the most frequent causes of occupational asthma, 2006–2015 (*n* = 1167) (European network for the PHenotyping of OCcupational Asthma). From data of Vandenplas et al. [32].

**Table 1 ijerph-17-04713-t001:** Distribution of 3253 putative agents by category reported for 2209 WEA cases identified by state-based surveillance programs in the United States, 1993–2006 and 2009–2012^1.^

Agent Category^2^	Examples of agents	No.	No. as % of 3253
Miscellaneous chemicals and materials	Chemicals n.o.s., perfume n.o.s., pesticides n.o.s., glues n.o.s.	523	16.1
Mineral and inorganic dusts	Dust n.o.s., cement dust, copier toner	477	14.7
Cleaning materials	Sodium hypochlorite, disinfectant cleaners, floor strippers, floor wax, carpet cleaners	440	13.5
Pyrolysis products	Cigarette smoke, diesel exhaust, plastic smoke	347	10.7
Indoor air pollutants	Indoor air pollutants, indoor air pollutants from renovations	241	7.4
Molds	Mold n.o.s.	179	5.5
Solvents, n.o.s.	Paint, lacquer, solvents n.o.s.	165	5.1
Plant materials	Capsicum, wood dust, pollen, flour	119	3.7
Ergonomics	Exercise, stress	98	3.0
Aliphatic and alicyclic hydrocarbons	4–Phenylcyclohexane, gasoline, petroleum fractions, n.o.s.	78	2.4
Acids, bases, and oxidizing agents	Sulfuric acid, hydrochloric acid, anhydrous ammonia, nitric acid, phosphoric acid	70	2.2
Animal materials	Cat, rat antigens, rat feces, insects n.o.s., animal dander n.o.s.	68	2.1
Physical factors	Cold and hot temperatures, high humidity	59	1.8
Metals and metalloids	Welding, metal dust n.o.s.	53	1.6
Other and unknown	n.a.	336	10.3
TOTAL	n.a.	3253	100

Abbreviations: n.a. = not applicable; n.o.s. = not otherwise specified; WEA: work-exacerbated asthma. ^1^ From sentinel surveillance data reported by programs in the states of California (CA), Massachusetts (MA), Michigan (MI), and New Jersey during 1993–2006 (*n* = 1307 WEA cases, 1925 agents) and in the states of CA, MA, MI, and New York during 2009–2012 (902 WEA cases, 1328 agents) [8]. Up to three putative agents could be reported for each case. Only agent categories with counts of 50 or greater are identified separately in the table, with the balance included in the final “Other and unknown” category. **^2^** The agent categories are based on the Association of Occupational and Environmental Clinics (AOEC) Exposure Code List as of April 2016. An explanation of the exposure list and access to it is available at http://www.aoec.org/tools.htm.

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
