# Peer review of "Causes and Phenotypes of Work-Related Asthma"

_ijerph, 2020, doi:10.3390/ijerph17134713_

Round 1

Reviewer 1 Report

The article appears to me to be a clear, straightforward, narrative review of recent articles in the field.

I have only a remark about the lay-out: 2. WEA has chapters (definition, prevalence, etc.). 3 & 4 have not. That is not consequent and could be interpreted as being of less importance...

Author Response

Reviewer 1

The article appears to me to be a clear, straightforward, narrative review of recent articles in the field. I have only a remark about the lay-out: 2. WEA has chapters (definition, prevalence, etc.). 3 & 4 have not. That is not consequent and could be interpreted as being of less importance...

Response: Thank you for the general comment. The subtitles of the 2. WEA paragraph have been deleted for consistency with the other paragraphs.

Reviewer 2 Report

General: The paper provides an updated review on the state on evidence of WPA, an important topic around the world.  The paper could benefit from more structure. Presenting individual studies without context (validity and generalizability, specific populations, country-specific differences in regulations, reporting, etc) makes it difficult to evaluate the quality of the evidence-base and the current state of knowledge. Study design should be emphasized (prospective, cross-sectional, surveillance reports) and populations surveyed should be mentioned.  A summary/conclusion at the end of each phenotype may increase clarity for the reader.

The conclusion reads much like the introduction and the abstract.  I suggest the authors speak to the strengths and weaknesses of the evidence-base.

Specific suggestions:

Abstract:  Present data to support the following statements:' additional proportion', 'relatively few categories', 'few agents'.

Introduction: Summarize significant advances that have been made that warrant a new review. 

Line 58: For the reader unfamiliar w/ ECRCHS, provide a brief description of the population and coverage. For example are the data from the US meant to represent the entire country?

Prevalance: Comment on the quality of the data supporting the finding of 21.5%. Is it based on valid and generalizable data in your opinion.  If there are limitations, please state them.

Line 114:  Quantify 'less likely'. Add a statement why the results are different. For example, could it be due to country-specific employment law differences.

Line 119:  Quality of life and functionality do not seem to go in that section.  There needs to be a segue to this topic. Quantify 'somewhat worse results' and 'similar scores'.  

Line 122: How do the findings on anxiety and depression compare to previous studies.

Line 138: As described, the Dumas study appears to be cross-sectional, in which case you can not conclude that exposure causes asthma. 

Line 145:  Unclear what the implications are of including patients with OA. A line explaining this would add clarity.

Line 151:  It would increase the  readers ability to interpret the information if more information was given. For example the type of studies, whether the cohorts were comparable, whether the same questionnaires used and the validity of them, were covariates controlled?

Line 162:  What was the working definition of respiratory disease, was it WPA?

'Attributed to' suggests caused by, is that the intention of the paragraph or was this cross-sectional data?

The conclusion could be strengthened. 

Author Response

Reviewer 2

General: The paper provides an updated review on the state on evidence of WPA, an important topic around the world.  The paper could benefit from more structure. Presenting individual studies without context (validity and generalizability, specific populations, country-specific differences in regulations, reporting, etc) makes it difficult to evaluate the quality of the evidence-base and the current state of knowledge. Study design should be emphasized (prospective, cross-sectional, surveillance reports) and populations surveyed should be mentioned. 

Response: Thank you for the general comments. We have taken your specific remarks and revised the manuscript accordingly

A summary/conclusion at the end of each phenotype may increase clarity for the reader.

Response: A conclusion at the end of each phenotype has been provided or highlighted.

The conclusion reads much like the introduction and the abstract.  I suggest the authors speak to the strengths and weaknesses of the evidence-base.

Response: The conclusions have been revised according with the suggestions

Specific suggestions:

Abstract:  Present data to support the following statements:' additional proportion', 'relatively few categories', 'few agents'.

Response: The Abstract has been revised as suggested

Introduction: Summarize significant advances that have been made that warrant a new review. 

Response: The revised Introduction anticipates what are the significant advances on WRA covered by the review, as suggested.

Line 58: For the reader unfamiliar w/ ECRCHS, provide a brief description of the population and coverage. For example are the data from the US meant to represent the entire country?

Response:  Thank you for the suggestion to add a brief description of the population and coverage.  Indeed, the data from the US represents a very small portion of the US population.  Changes to the text are indicated in red.

A fourth study was based on the European Community Respiratory Health Survey (ECRHS) and used data from 25 study centers in 11  European countries and one center in the US [5].  Participants had been selected either at random or because they had reported respiratory symptoms during the first round of the ECRHS (1991-1993) and had also taken part in the second round (1998-2003).

Prevalence: Comment on the quality of the data supporting the finding of 21.5%. Is it based on valid and generalizable data in your opinion.  If there are limitations, please state them.

Response:  We made the following modifications and additions at the end of the paragraph:

The median WEA prevalence among for the combination of these more recent studies and the prior 12 was 21.5%, the same as reported in 2011. A potential limitation of most of these studies was an inclusive definition of WEA based on a self-reported association of asthma symptoms with work. Three studies referenced in the 2011 ATS statement used more objective criteria for WEA and had a lower median prevalence estimate of 14% [1]. One of the more recent studies applied more objective criteria for WEA and reported a 15% prevalence [4].  The four studies considered together had a median estimate of 14.5%.

Line 114:  Quantify 'less likely'. Add a statement why the results are different. For example, could it be due to country-specific employment law differences.

Response to first part of comment:  We quantified “less likely” by making the following changes:

In confirmation of the earlier observations, both studies reported that patients with WEA and OA were equally likely to be employed at follow-up, and the Quebec study found that job changes were less likely with WEA (42%) than OA (77%).  Even after adjusting for potential confounders, the WEA patients in Quebec were more likely to have kept a job with the same employer than their OA counterparts [11].

Response to second part of comment:  Results for income loss in the 2011 ATS document were from studies conducted in the United Kingdom and Belgium. Therefore, the contrast with more recent findings could possibly be influenced by country-specific differences in provisions for the different work-related asthma cases. 

However, the results for income loss differed from those presented in the ATS statement.  Specifically, income loss was less common for WEA than OA in the Ontario study [10], and any change in income of at least $5000 was less common for WEA in the Quebec study [11].  Results for income loss in the ATS statement were from studies conducted in the United Kingdom and Belgium and showed little difference between the two types of cases. The contrast with findings from Canada could be influenced by country-specific compensation and employment law differences.

Line 119:  Quality of life and functionality do not seem to go in that section.  There needs to be a segue to this topic. Quantify 'somewhat worse results' and 'similar scores'.  

Response:  After reviewing the text, we agree it is a bad fit for this section entitled “Socioeconomic factors.”  Because of this and the addition of new text in this section, we decided to delete the sentences in question:

In addition, the two Canadian studies reported that WEA and OA cases had similar results for indicators of functionality, specifically for quality of life and work limitations.  For psychological status, the Ontario study noted a trend towards somewhat worse results on indexes of anxiety and depression for WEA cases, but the contrasts were not statistically significant [10].  The Quebec study reported that WEA and OA had similar scores for both anxiety and depression [11].  

Line 122: How do the findings on anxiety and depression compare to previous studies.

Response:  Please see the response to the previous comment.  The 2011 ATS statement included no results on anxiety and depression.

Line 138: As described, the Dumas study appears to be cross-sectional, in which case you can not conclude that exposure causes asthma. 

Response: We agree. The reviewer is correct that the study by Dumas and colleagues (line 140) was cross-sectional and does not prove causation. In the paragraph that includes this study, the preceding sentence is worded to indicate that the subsequent studies provide some support that low-dose /moderate irritant exposure MAY cause asthma. We have not asserted that this is proven and the studies in this section have all reflected associations rather than proof of causation. Nevertheless they provide some level of support for this concept.

Line 145:  Unclear what the implications are of including patients with OA. A line explaining this would add clarity.

Response: Thank you for this comment. We agree that it is difficult from these studies to determine the mechanism of asthma and have added a sentence to this effect.

Line 151:  It would increase the readers ability to interpret the information if more information was given. For example the type of studies, whether the cohorts were comparable, whether the same questionnaires used and the validity of them, were covariates controlled?

Response: some additional information has now been added as to the type of study and cohorts, but due to word count limitations we have not included full details and would expect that the interested reader would obtain the full papers to assess these further. Reference 20 and 21 (Dumas et al) are from the same initial cohort (Nurses’ study) and the same time period but have slightly different entry criteria for analyses and number of subjects therefore differ. Covariates were included in analyses.

Line 162:  What was the working definition of respiratory disease, was it WPA? 'Attributed to' suggests caused by, is that the intention of the paragraph or was this cross-sectional data?

Response: lines 164,165: this paper analyzed all cases that were reported by physicians to an occupational surveillance system in the time period stated, because of suspected work-related respiratory disease following exposure to cleaning agents. A case could be reported by a primary care physician or specialist even if there was no objective proof of causation and the term “attributed” refers to the attribution made by the reporting physician based on the reporting of symptoms by the worker, occurring after exposure and potentially in some cases supported by further tests – but as we had indicated, clinical details were not provided. The majority of cases reported in this paper were from chest physicians and 60% of these were given a diagnosis of asthma as was indicated, but “inhalation accident” and “other respiratory disease” were also common.

The conclusion could be strengthened.

Response: Please, see above under general comment

Reviewer 3 Report

Sir, than you for the opportunity to review this very interesting paper about "Causes and phenotypes of work-related asthma". Authors, all of them researchers with a very high profile, all of them with a deep understanding of this topic, and having previously participated to the panel that contributed to the case definition of WRA, have reported a narrative review on "work-related astma". The text is properly written, organized, and up-to-date.

I've only the following (minor) recommendations:

a) as 3D graph may be misleading, change in 2D Figure 1, formatting accordingly with the editorial recommendations its legend;

b) change the last 2 cells from the first row of table 1 as follows 

from "N" to "No./3253" and from "Percentage of 3253" to "%"

Author Response

Reviewer 3

Sir, than you for the opportunity to review this very interesting paper about "Causes and phenotypes of work-related asthma". Authors, all of them researchers with a very high profile, all of them with a deep understanding of this topic, and having previously participated to the panel that contributed to the case definition of WRA, have reported a narrative review on "work-related astma". The text is properly written, organized, and up-to-date.I've only the following (minor) recommendations:

  1. a) as 3D graph may be misleading, change in 2D Figure 1, formatting accordingly with the editorial recommendations its legend;

Response: The content of each section of the 3D graph is clearly specified (agent and %). Therefore, we do not believe that the Figure 1 has elements to be misleading and prefer to maintain the 3D graph. Title and caption have been made according with the Instruction for Author of IJERPH.

  1. b) change the last 2 cells from the first row of table 1 as follows from "N" to "No./3253" and from "Percentage of 3253" to "%"

Response:  The suggestion for the next to last cell in the first row of Table 1 – No./3253 – would misrepresent the content of the column.  That column only contains “No.” and not “No./3253.”  It appears the reviewer considers it undesirable to use “N,” so we changed it to “No.” We changed the last cell from “Percentage of 3253" to “No. as % of 3253" to be consistent with the change in the neighboring cell and to more clearly communicate the content of the final column.

Reviewer 4 Report

Very well written and presented, I made 2 minor editorial suggestions on page 2 line 50 and 59-60. See attached.

Author Response

Reviewer 4

Very well written and presented, I made 2 minor editorial suggestions on page 2 line 50 and 59-60. See attached.

Response: Thank you for the general comment. The suggestions have been incorporated in the revised manuscript.

Round 2

Reviewer 2 Report

The authors responsed to all of my concerns about the previous version.